Extended Abstract Track

# Expander Graph Propagation

**Andreea Deac**                                                    DEACANDR@MILA.QUEBEC
*Mila, Université de Montréal*

**Marc Lackenby**                                                  LACKENBY@MATHS.OX.AC.UK
*University of Oxford*

**Petar Veličković**                                              PETARV@DEEPMIND.COM
*DeepMind*

**Editors:** Sophia Sanborn, Christian Shewmake, Simone Azeglio, Arianna Di Bernardo, Nina Miolane

## Abstract

Deploying graph neural networks (GNNs) on whole-graph classification or regression tasks is challenging, often requiring node features that are mindful of both local interactions and the graph global context. GNN architectures need to avoid pathological behaviours, such as bottlenecks and oversquashing, while ideally having linear time and space complexity requirements. In this work, we propose an elegant approach based on propagating information over *expander graphs*. We provide an efficient method for constructing expander graphs of a given size, and use this insight to propose the EGP model. We show that EGP is able to address all of the above concerns, while requiring minimal effort to set up, and provide evidence of its empirical utility on relevant datasets and baselines in the Open Graph Benchmark. Importantly, using expander graphs as a template for message passing necessarily gives rise to negative curvature. While this appears to be counterintuitive in light of recent related work on oversquashing, we theoretically demonstrate that negatively curved edges are likely to be **required** to obtain scalable message passing without bottlenecks.

**Keywords:** graph neural networks, graph representation learning, graph machine learning, bottlenecks, oversquashing, curvature, expander graphs, cayley graphs, group theory

## 1. Introduction

Most GNNs rely on propagating information between neighbouring nodes in the graph (Bronstein et al., 2021). However, in many areas of scientific interest, purely local interactions are likely to be insufficient, while merely stacking more message passing layers over the input graph is often inadequate, as it leaves GNNs vulnerable to pathological behaviours such as oversquashing (Alon and Yahav, 2020), wherein nodes close to *bottlenecks* in the graph would need to store quantities of information that are *exponentially* increasing with depth.

Precisely, we seek a method that satisfies *four* desirable criteria: (**C1**) it is capable of propagating information *globally* in the graph; (**C2**) it is *resistant* to the oversquashing effect and does not introduce bottlenecks; (**C3**) its time and space complexity remain *subquadratic*; and (**C4**) it requires *no dedicated preprocessing* of the input. We survey prior methods in detail within Appendix A, demonstrating they do not accomplish (**C1**–**C4**) simultaneously.

In this paper, we identify *expander graphs* as very attractive objects in this regard. Specifically, they offer a family of graph structures that are fundamentally *sparse*, while having *low diameter*: thus, any two nodes in an expander graph may reach each other in a short number of hops, eliminating bottlenecks (Figure 1). Further, we will demonstrate an

Extended Abstract Track

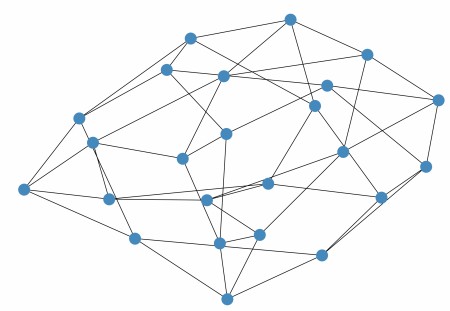 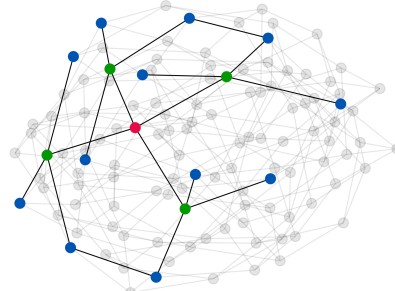

Figure 1: **Left**: The Cayley graph of $\mathrm{SL}(2, \mathbb{Z}_3)$. It has $|V| = 24$ and it is 4-regular. Despite its sparsity, it is highly interconnected, with a diameter of 4. **Right**: The Cayley graph of $\mathrm{SL}(2, \mathbb{Z}_5)$, with $|V| = 120$. A 2-hop neighbourhood of a node (red) is highlighted, demonstrating its tree-like local structure.

efficient way to construct a family of expander graphs (leveraging known results on the *special linear group*, $\mathrm{SL}(2, \mathbb{Z}_n)$). Once an expander graph of appropriate size is constructed, we perform a certain number of GNN *propagation* steps over its structure to globally distribute the nodes' features. Accordingly, we name our method *expander graph propagation* (**EGP**).

We also show that, in spite of their negative curvature, our expander graphs never trigger the conditions necessary for the oversquashing results in Topping et al. (2021) to be applicable, and prove that the existence of negatively curved edges might in fact be **required** in order to have sparse communication without bottlenecks.

## 2. Theoretical background

**Definition 1** *A collection $\{G_i\}$ of finite connected graphs is an* expander family *if there is a constant $c > 0$ such that for all $G_i$ in the collection, $\lambda_1(G_i) \geqslant c$, where $\lambda_1(G_i)$ is the second-smallest eigenvalue of the graph Laplacian of $G_i$.*

**Definition 2** *Let $G$ be a finite graph. For $A \subset V(G)$, its* boundary $\partial A$ *is the collection of edges connecting a node in $A$ to a node not in $A$. The* Cheeger constant $h(G)$ *is defined as*

$$h(G) = \min \left\{ \frac{|\partial A|}{|A|} : A \subset V(G), 0 < |A| \leqslant |V(G)|/2 \right\}.$$

Having a small Cheeger constant is equivalent to the graph having a 'bottleneck'. Expander families can be reinterpreted using Cheeger constants, as follows (see, e.g., Alon (1986); Alon and Milman (1985); Dodziuk (1984); Tanner (1984)):

**Theorem 3** *Let $\{G_i\}$ be a collection of finite connected graphs with a uniform upper bound on their vertex degrees. Then $\{G_i\}$ is an expander family iff there is a constant $\epsilon > 0$ such that for all graphs in the collection, $h(G_i) \geqslant \epsilon$.*

Hence, expanders have higher Cheeger constants and will hence be *bottleneck-free*. We also show expanders have favourable *diameter*; see Mohar (1991, Theorem 2.3) for a proof.

Extended Abstract Track

**Theorem 4** *The diameter* $\mathrm{diam}(G)$ *of a graph* $G$ *satisfies*

$$\mathrm{diam}(G) \leqslant 2 \left\lceil \frac{\Delta(G) + \lambda_1(G)}{4\lambda_1(G)} \log(|V(G)| - 1) \right\rceil,$$

*where* $\Delta(G)$ *is the maximal degree of any vertex of* $G$. *Hence, if* $\{G_i\}$ *is an expander family of finite graphs with a uniform upper bound on their vertex degrees, then there is a constant* $k > 0$ *such that for all graphs in the family,* $\mathrm{diam}(G_i) \leqslant k \log V(G_i)$.

Therefore, globally propagating information over an expander with $|V|$ nodes only requires $O(\log |V|)$ propagation steps—yielding subquadratic complexity. To efficiently construct expanders of (roughly) $|V|$ nodes, we leverage results from group theory (Appendix B).

**Definition 5** *Let* $\Gamma$ *be a group with a finite generating set* $S$. *Then the associated* Cayley graph $\mathrm{Cay}(\Gamma; S)$ *has vertex set* $\Gamma$ *and it has an edge* $g \to gs$ *for each* $g \in \Gamma$ *and each* $s \in S$.

The degree of each vertex of a Cayley graph $\mathrm{Cay}(\Gamma; S)$ is $2|S|$. This is because each vertex $g$ is joined by edges to $gs$ and $gs^{-1}$ for each $s \in S$. To preserve sparsity, we are interested in the case where there is a uniform upper bound on $|S|$; the key example follows.

For $n \in \mathbb{N}$, the *special linear group* $\mathrm{SL}(2, \mathbb{Z}_n)$ denotes the group of $2 \times 2$ matrices with entries that are integers modulo $n$ and with determinant 1. One of its generating sets is $S_n = \left\{ \begin{pmatrix} 1 & 1 \\ 0 & 1 \end{pmatrix}, \begin{pmatrix} 1 & 0 \\ 1 & 1 \end{pmatrix} \right\}$. Central to our constructions is the following result, for which proofs are given in Kowalski (2019); Davidoff et al. (2003), using result of Selberg (1965):

**Theorem 6** *The family of Cayley graph* $\mathrm{Cay}(\mathrm{SL}(2, \mathbb{Z}_n); S_n)$ *forms an expander family.*

## 3. Local structure of Cayley graphs, and the utility of negative curvature

Recent work (Topping et al., 2021) suggested that the local structure of the graph $G$ used by a GNN plays a major part in the oversquashing effect. Specifically, *negatively curved edges*—using either the (balanced) Forman curvature (Forman, 2003) or Ollivier curvature (Ollivier, 2007, 2009)—were shown to be the culprits. Firstly, and surprisingly, we prove in Appendix C that edges in $G_n$ are *never* positively curved, with curvatures as low as $-1$.

However, further qualifying the findings in Topping et al. (2021), we contend that negative Ricci curvature is not in itself an impediment to efficient propagation around a GNN. It was shown in Topping et al. (2021, Theorem 4) that poor propagation arises when the balanced Forman curvature is close to $-2$, specifically if it is at most $-2 + \delta$; however, with certainty, $\delta = 1$ can *never* be satisfied in the hypotheses of Topping et al. (2021, Theorem 4).

Furthermore, positive Ricci curvature may have *downsides* when used for GNNs. Using the main result of Salez (2021), we can show that the three properties of expansion, sparsity and non-negative Ollivier curvature are incompatible, in the following sense.

**Theorem 7** *For any* $\delta > 0$ *and* $\Delta > 0$, *there are only finitely many graphs with maximum vertex degree* $\Delta$, *Cheeger constant at least* $\delta$ *and non-negative Ollivier curvature.*

Table 1: Comparative evaluation on the four datasets studied.

| Model | ogbg-molhiv | ogbg-molpcba | ogbg-ppa | ogbg-code2 |
|---|---|---|---|---|
| GIN | $0.7558 \pm 0.0140$ | $0.2266 \pm 0.0028$ | $0.6892 \pm 0.0100$ | $0.1495 \pm 0.0023$ |
| GIN + EGP | $\mathbf{0.7934} \pm 0.0035$ | $\mathbf{0.2329} \pm 0.0019$ | $\mathbf{0.7027} \pm 0.0159$ | $\mathbf{0.1497} \pm 0.0015$ |

We prove Theorem 7 in Appendix E. In our view, it is highly desirable that graphs used for GNNs have high Cheeger constants, in the sense of globally lacking bottlenecks. Having bounded degree is useful too, as graphs will be sparse, and the nodes will not have to handle larger neighbourhoods as graphs grow larger in size. As we have just shown, non-negative Ollivier curvature is *incompatible* with these properties when the graph is sufficiently large.

## 4. Expander graph propagation

Our GNN input contains a node feature matrix $\mathbf{X} \in \mathbb{R}^{|V| \times k}$, and an adjacency matrix $\mathbf{A} \in \mathbb{R}^{|V| \times |V|}$. We ignore edge features for simplicity, without changing our findings.

The crux of our method is leveraging the computed Cayley graph $\mathrm{Cay}(\mathrm{SL}(2, \mathbb{Z}_n); S_n)$ for message propagation. We opt for a simple construction: interleave running a standard GNN over the given input structure, followed by running another GNN layer over the Cayley graph. If $\mathbf{A}^{\mathrm{Cay}(n)}$ is the adjacency matrix derived from $\mathrm{Cay}(\mathrm{SL}(2, \mathbb{Z}_n); S_n)$, this implies $\mathbf{H} = \mathrm{GNN}(\mathrm{GNN}(\mathbf{X}, \mathbf{A}), \mathbf{A}^{\mathrm{Cay}(n)})$. Here, we use the GIN (Xu et al., 2018) as our GNN:

$$\mathbf{h}_u = \phi \left( (1 + \epsilon) \mathbf{x}_u + \sum_{v \in \mathcal{N}_u} \mathbf{x}_v \right) \tag{1}$$

where $\mathcal{N}_u$ is the neighbourhood of node $u$, i.e. in our setup, the set of all nodes $v$ such that $a_{vu} \neq 0$. $\epsilon \in \mathbb{R}$ is a learnable scalar, and $\phi \colon \mathbb{R}^k \to \mathbb{R}^{k'}$ is a two-layer MLP.

We assumed the number of nodes in our input graph to line up with the Cayley graph, that is, $\mathbf{A}^{\mathrm{Cay}(n)} \in \mathbb{R}^{|V| \times |V|}$. However, there is no guarantee that $n \in \mathbb{N}$ exists, such that $\mathrm{Cay}(\mathrm{SL}(2, \mathbb{Z}_n); S_n)$ has $|V|$ nodes. As an approximation, we choose the smallest $n$ such that the number of nodes of $\mathrm{Cay}(\mathrm{SL}(2, \mathbb{Z}_n); S_n)$ is $\geq |V|$, then consider $\mathbf{A}^{\mathrm{Cay}(n)}_{1:|V|, 1:|V|}$—i.e. only the subgraph containing the first $|V|$ nodes in the Cayley graph. Further, in all our experiments we construct the Cayley graph in a breadth-first manner, starting from the identity element as "node zero"—this guarantees that we do not disconnect the graph by taking this subgraph. Appendix I summarises the steps of our proposed EGP model in pseudocode.

## 5. Empirical evaluation

We provide direct comparative experiments in order to supplement our theory, and ascertain that EGP can directly help existing graph classifiers, even without extensive tuning. We leverage the Open Graph Benchmark (Hu et al., 2020, OGB) graph classification datasets: `ogbg-molhiv`, `ogbg-molpcba`, `ogbg-ppa` and `ogbg-code2`.

In all four datasets, we want to *directly* evaluate the empirical gain of introducing an EGP layer and completely rule out any effects from parameter count, or similar architectural

Extended Abstract Track

decisions. Our baseline model is the GIN (Xu et al., 2018), with hyperparameters as given by (Hu et al., 2020). We use the *official* publicly available model implementation from the OGB authors (Hu et al., 2020), and modify all *even* layers of the architecture to operate over the appropriately-sampled Cayley graph.

Note that our construction leaves both the parameter count and latent dimension of the model *unchanged*, hence any benefits coming from optimising those have been diminished.

The results of our evaluation are presented in Table 1. It can be observed that, in all four cases, propagating information over the Cayley graph yields improvements in mean performance—these improvements are most apparent on `ogbg-molhiv`. We believe that these results provide encouraging empirical evidence that propagating information over Cayley graphs is an elegant idea for alleviating bottlenecks.

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

# Extended Abstract Track

Michael M Bronstein, Joan Bruna, Taco Cohen, and Petar Veličković. Geometric deep learning: Grids, groups, graphs, geodesics, and gauges. *arXiv preprint arXiv:2104.13478*, 2021.

Fan R. K. Chung. *Spectral graph theory*, volume 92 of *CBMS Regional Conference Series in Mathematics*. Published for the Conference Board of the Mathematical Sciences, Washington, DC; by the American Mathematical Society, Providence, RI, 1997. ISBN 0-8218-0315-8.

Giuliana Davidoff, Peter Sarnak, and Alain Valette. *Elementary number theory, group theory, and Ramanujan graphs*, volume 55 of *London Mathematical Society Student Texts*. Cambridge University Press, Cambridge, 2003. ISBN 0-521-82426-5; 0-521-53143-8. doi: 10.1017/CBO9780511615825. URL https://doi.org/10.1017/CBO9780511615825.

Jozef Dodziuk. Difference equations, isoperimetric inequality and transience of certain random walks. *Trans. Amer. Math. Soc.*, 284(2):787–794, 1984. ISSN 0002-9947. doi: 10.2307/1999107. URL https://doi.org/10.2307/1999107.

Matthias Fey, Jan-Gin Yuen, and Frank Weichert. Hierarchical inter-message passing for learning on molecular graphs. *arXiv preprint arXiv:2006.12179*, 2020.

R Forman. Discrete and computational geometry. 2003.

Johannes Gasteiger, Stefan Weißenberger, and Stephan Günnemann. Diffusion improves graph learning. *arXiv preprint arXiv:1911.05485*, 2019.

Justin Gilmer, Samuel S Schoenholz, Patrick F Riley, Oriol Vinyals, and George E Dahl. Neural message passing for quantum chemistry. In *International conference on machine learning*, pages 1263–1272. PMLR, 2017.

Aditya Grover and Jure Leskovec. node2vec: Scalable feature learning for networks. In *Proceedings of the 22nd ACM SIGKDD international conference on Knowledge discovery and data mining*, pages 855–864, 2016.

Weihua Hu, Matthias Fey, Marinka Zitnik, Yuxiao Dong, Hongyu Ren, Bowen Liu, Michele Catasta, and Jure Leskovec. Open graph benchmark: Datasets for machine learning on graphs. *Advances in neural information processing systems*, 33:22118–22133, 2020.

Emmanuel Kowalski. *An introduction to expander graphs*, volume 26 of *Cours Spécialisés [Specialized Courses]*. Société Mathématique de France, Paris, 2019. ISBN 978-2-85629-898-5.

Devin Kreuzer, Dominique Beaini, Will Hamilton, Vincent Létourneau, and Prudencio Tossou. Rethinking graph transformers with spectral attention. *Advances in Neural Information Processing Systems*, 34, 2021.

Johannes F Lutzeyer, Changmin Wu, and Michalis Vazirgiannis. Sparsifying the update step in graph neural networks. *arXiv preprint arXiv:2109.00909*, 2021.

# Extended Abstract Track

Grigorii A Margulis. Explicit constructions of graphs without short cycles and low density codes. *Combinatorica*, 2(1):71–78, 1982.

Grégoire Mialon, Dexiong Chen, Margot Selosse, and Julien Mairal. Graphit: Encoding graph structure in transformers. *arXiv preprint arXiv:2106.05667*, 2021.

Bojan Mohar. Eigenvalues, diameter, and mean distance in graphs. *Graphs Combin.*, 7(1): 53–64, 1991. ISSN 0911-0119. doi: 10.1007/BF01789463. URL https://doi.org/10.1007/BF01789463.

Christopher Morris, Martin Ritzert, Matthias Fey, William L Hamilton, Jan Eric Lenssen, Gaurav Rattan, and Martin Grohe. Weisfeiler and leman go neural: Higher-order graph neural networks. In *Proceedings of the AAAI conference on artificial intelligence*, volume 33, pages 4602–4609, 2019.

Christopher Morris, Gaurav Rattan, and Petra Mutzel. Weisfeiler and leman go sparse: Towards scalable higher-order graph embeddings. *Advances in Neural Information Processing Systems*, 33:21824–21840, 2020.

Yann Ollivier. Ricci curvature of metric spaces. *Comptes Rendus Mathematique*, 345(11): 643–646, 2007.

Yann Ollivier. Ricci curvature of markov chains on metric spaces. *Journal of Functional Analysis*, 256(3):810–864, 2009.

Bryan Perozzi, Rami Al-Rfou, and Steven Skiena. Deepwalk: Online learning of social representations. In *Proceedings of the 20th ACM SIGKDD international conference on Knowledge discovery and data mining*, pages 701–710, 2014.

Justin Salez. Sparse expanders have negative curvature. *arXiv preprint arXiv:2101.08242*, 2021.

Adam Santoro, David Raposo, David G Barrett, Mateusz Malinowski, Razvan Pascanu, Peter Battaglia, and Timothy Lillicrap. A simple neural network module for relational reasoning. *Advances in neural information processing systems*, 30, 2017.

Atle Selberg. On the estimation of Fourier coefficients of modular forms. In *Proc. Sympos. Pure Math., Vol. VIII*, pages 1–15. Amer. Math. Soc., Providence, R.I., 1965.

Kimberly Stachenfeld, Jonathan Godwin, and Peter Battaglia. Graph networks with spectral message passing. *arXiv preprint arXiv:2101.00079*, 2020.

Jian Tang, Meng Qu, Mingzhe Wang, Ming Zhang, Jun Yan, and Qiaozhu Mei. Line: Large-scale information network embedding. In *Proceedings of the 24th international conference on world wide web*, pages 1067–1077, 2015.

R. Michael Tanner. Explicit concentrators from generalized $N$-gons. *SIAM J. Algebraic Discrete Methods*, 5(3):287–293, 1984. ISSN 0196-5212. doi: 10.1137/0605030. URL https://doi.org/10.1137/0605030.

# Extended Abstract Track

Shantanu Thakoor, Corentin Tallec, Mohammad Gheshlaghi Azar, Mehdi Azabou, Eva L Dyer, Remi Munos, Petar Veličković, and Michal Valko. Large-scale representation learning on graphs via bootstrapping. In *International Conference on Learning Representations*, 2021.

Jake Topping, Francesco Di Giovanni, Benjamin Paul Chamberlain, Xiaowen Dong, and Michael M Bronstein. Understanding over-squashing and bottlenecks on graphs via curvature. *arXiv preprint arXiv:2111.14522*, 2021.

Jean-Pierre Serre Trees. Translated from the french original by john stillwell. corrected 2nd printing of the 1980 english translation. *Springer Monographs in Mathematics. Springer-Verlag, Berlin*, 2003.

Petar Veličković, William Fedus, William L Hamilton, Pietro Liò, Yoshua Bengio, and R Devon Hjelm. Deep graph infomax. *arXiv preprint arXiv:1809.10341*, 2018.

Keyulu Xu, Weihua Hu, Jure Leskovec, and Stefanie Jegelka. How powerful are graph neural networks? *arXiv preprint arXiv:1810.00826*, 2018.

Chengxuan Ying, Tianle Cai, Shengjie Luo, Shuxin Zheng, Guolin Ke, Di He, Yanming Shen, and Tie-Yan Liu. Do transformers really perform badly for graph representation? *Advances in Neural Information Processing Systems*, 34, 2021.

# Extended Abstract Track

Table 2: A summary of principal approaches to handling global context in graph representation learning (Section A). "(✔)" indicates that a criterion *may* be satisfied, depending on the method's tradeoffs. Our proposal, the expander graph propagation (EGP) method, satisfies all four criteria.

| Approach | (**C1**) (global prop.) | (**C2**) (no bottlenecks) | (**C3**) (subquadratic) | (**C4**) (no dedicated preproc.) |
|---|:---:|:---:|:---:|:---:|
| GNNs | ✗ | ✗ | ✓ | ✓ |
| Sufficiently deep GNNs | ✓ | ✗ | ✗ | ✓ |
| Master node | ✓ | ✗ | ✓ | ✓ |
| Fully connected | ✓ | ✓ | ✗ | ✓ |
| Feature aug. | ✓ | (✓) | (✓) | ✗ |
| Graph rewiring | ✓ | ✓ | ✓ | ✗ |
| Hierarchical MP | ✓ | ✓ | (✓) | ✗ |
| **EGP** (ours) | ✓ | ✓ | ✓ | ✓ |

## Appendix A. An in-depth analysis of prior art

In this appendix, we survey many of prior approaches to handling global context in graph representation learning, evaluating them carefully against our four desirable criteria (**C1**–**C4**; cf. Table 2). This list is by no means exhaustive, but should be indicative of the most important directions.

**Stacking more layers.** As already highlighted, one way to achieve global information propagation is to have a deeper GNN. In this case, we are capable of satisfying (**C1**) and (**C4**)—no dedicated preprocessing is needed. However, depending on the graph's diameter, we may need up to $O(|V|)$ layers to cover the graph, leading to quadratic complexity (violating (**C3**)) and introducing a vulnerability to bottlenecks (**C2**), as theoretically and empirically demonstrated in (Alon and Yahav, 2020).

**Master nodes.** An attractive approach to introducing global context is to introduce a *master node* to the graph, and connect it to all of the graph's nodes. This can be done either explicitly (Gilmer et al., 2017) or implicitly, by storing a "global" vector (Battaglia et al., 2018). It trivially reduces the graph's diameter to 2, introduces $O(1)$ new nodes and $O(|V|)$ new edges, and requires no dedicated preprocessing, hence it satisfies (**C1**, **C3**, **C4**). However, these benefits come at the expense of introducing a bottleneck in the master node: it has a very challenging task (especially when graphs get larger) to continually incorporate information over a very large neighbourhood in a useful way. Hence it fails to satisfy (**C2**).

**Fully connected graphs.** The converse approach is to make *every* node a master node: in this case, we make all pairs of nodes connected by an edge—this was initially proposed as a powerful method to alleviate oversquashing by (Alon and Yahav, 2020). This strategy proved highly popular in the recent surge of Graph Transformers (Kreuzer et al., 2021; Mialon et al., 2021; Ying et al., 2021), and is common for GNNs used in physical simulation (Battaglia et al., 2016) or reasoning (Santoro et al., 2017) tasks. The graph's diameter is reduced to 1, no bottlenecks remain, and the approach does not require any dedicated preprocessing.

# *Extended Abstract Track*

Hence (**C1**, **C2**, **C4**) are trivially satisfied. The main downside of this approach is the introduction of $O(|V|^2)$ edges, which means (**C3**) can never be satisfied—and this approach will hence be prohibitive even for modestly-sized graphs.

**Feature augmentation.** An alternative approach is to provide additional features to the GNN which directly identify the structural role each node plays in the graph structure (Bouritsas et al., 2022). If done properly (i.e., if the computed features are directly relevant to the target task), this can drastically improve expressive power. Hence, in theory, it is possible to satisfy (**C1**) while not violating (**C2**, **C3**). However, computing appropriate features requires either specific domain knowledge, or an appropriate pre-training procedure (Grover and Leskovec, 2016; Perozzi et al., 2014; Tang et al., 2015; Thakoor et al., 2021; Veličković et al., 2018) to be applied, in order to obtain such embeddings. Hence all of these gains come at the expense of failing to satisfy (**C4**).

**Graph rewiring.** Another promising line of research involves modifying the edges of the original graph, in order to alleviate bottlenecks. Popular examples of this approach involve using diffusion (Gasteiger et al., 2019)—which diffuse additional edges through the application of kernels such as the personalised PageRank, and stochastic discrete Ricci flows (Topping et al., 2021)—which surgically modify a small quantity of edges to alleviate the oversquashing effect on the nodes with negative Ricci curvature. If realised carefully, such approaches will not deviate too far from the original graph, while provably alleviating oversquashing; hence it is possible to satisfy (**C1**, **C2**, **C3**). However, this comes at a cost of having to examine the input graph structure, with methods that do not necessarily scale easily with the number of nodes. As such, dedicated preprocessing is needed, failing to satisfy (**C4**).

**Hierarchical message passing.** Lastly, going beyond modifying the edges, it is also possible to introduce additional *nodes* in the graph—each of them responsible for a particular *substructure* in the graph[1]. If done carefully, it has the potential to drastically reduce the graph's diameter while not introducing bottlenecked nodes (hence, allowing us to satisfy (**C1**, **C2**)). However, in prior work, a cost has to be paid for this, usually in the need for dedicated preprocessing. Prior proposals for hierarchical GNNs that remain scalable require a dedicated pre-processing step (Bodnar et al., 2021a,b; Fey et al., 2020), sometimes coupled with domain knowledge (Fey et al., 2020)—thus failing to satisfy (**C4**). In addition, such methods may require adding prohibitively large numbers of substructures (Morris et al., 2020, 2019) or expensive pre-computation, e.g. computing the graph Laplacian eigenvectors (Stachenfeld et al., 2020). This might make even (**C3**) hard to satisfy.

We remark that our work is not the first to study expander graph-related topics in the context of GNNs. Specifically, the ExpanderGNN (Lutzeyer et al., 2021) leverages expander graphs over neural network weights to sparsify the update step in GNNs, and the Cheeger constant has been previously used to quantify oversquashing in (Topping et al., 2021). With respect to our contributions, neither of these cases discuss expander graphs in the context of the computational graph for a GNN, nor attempt to propagate messages over such a structure. Further, neither of these proposals successfully satisfies all four of our desired criteria (**C1**–**C4**).

---

1. The master node approach discussed before is a special case of this, wherein a single node is responsible for a "substructure" spanning the entire graph.

*Extended Abstract Track*

## Appendix B. Extended theoretical background

**Definition 8** *For a finite connected graph $G = (V(G), E(G))$, we consider functions $f: V(G) \to \mathbb{R}$. The* Laplacian *$Lf: V(G) \to \mathbb{R}$ of such a function is defined to be*

$$Lf(v) = \deg(v)f(v) - \sum_{vw \in E(G)} f(w),$$

*where $\deg(v)$ is the degree of the vertex $v$.*

The mapping $L: \mathbb{R}^{V(G)} \to \mathbb{R}^{V(G)}$ sending a function $f$ to its Laplacian $Lf$ is a linear transformation. It is not hard to show Chung (1997) that $L$ is symmetric with respect to the standard basis for $\mathbb{R}^{V(G)}$ and positive semi-definite and hence has non-negative real eigenvalues

$$0 = \lambda_0(G) < \lambda_1(G) \leqslant \lambda_2(G) \leqslant \dots.$$

The smallest eigenvalue is 0 and its associated eigenspace consists of the constant functions (assuming $G$ is connected). The smallest positive eigenvalue, $\lambda_1(G)$, is central to the definition of expander graphs, as the next definition shows.

**Definition 9** *A group $(\Gamma, \circ)$ is a set $\Gamma$ equipped with a* composition *operation $\circ: \Gamma \times \Gamma \to \Gamma$ (written concisely by omitting $\circ$, i.e. $g \circ h = gh$, for $g, h \in \Gamma$), satisfying the following axioms:*

- (Associativity) $(gh)l = g(hl)$, for $g, h, l \in \Gamma$.

- (Identity) *There exists a unique $e \in \Gamma$ satisfying $eg = ge = g$ for all $g \in \Gamma$.*

- (Inverse) *For every $g \in \Gamma$ there exists a unique $g^{-1} \in \Gamma$ such that $gg^{-1} = g^{-1}g = e$.*

**Definition 10** *Let $\Gamma$ be a group. A subset $S \subseteq \Gamma$ is a* generating set *for $\Gamma$ if it can be used to "generate" all of $\Gamma$ via composition. Concretely, any element $g \in \Gamma$ can be expressed by composing elements in the generating set, or their inverses; that is, we can express $g = s_1^{\pm 1} s_2^{\pm 1} s_3^{\pm 1} \cdots s_{n-1}^{\pm 1} s_n^{\pm 1}$ for $s_i \in S$.*

## Appendix C. Cayley graphs *never* have positive curvature

The various notions of curvature we discussed are defined for each $e$ of the graph $G$. Since, as defined by (Topping et al., 2021), the balanced Forman curvature of an edge depends only on local structures (i.e. triangles and squares) around that edge, they can be determined by only observing the immediate 2-hop surrounding of that edge. Formally, for an edge $e$ of a graph $G$, let $N_2(e)$ be the induced subgraph with vertices that are at most two hops away from at least one endpoint of $e$. Then the curvature of $e$ only depends on the isomorphism type of $N_2(e)$. More specifically, if $e$ and $e'$ are edges in possibly distinct graphs, and there is a graph isomorphism between $N_2(e)$ and $N_2(e')$ that sends $e$ to $e'$, then this guarantees that the curvatures of $e$ and $e'$ are equal.

This situation arises prominently in the Cayley graphs that we are considering, as follows.

Extended Abstract Track

**Proposition 11**  *Let s be one of*

$$\begin{pmatrix} 1 & 1 \\ 0 & 1 \end{pmatrix}, \qquad \begin{pmatrix} 1 & 0 \\ 1 & 1 \end{pmatrix}.$$

*Let $n, n' > 18$ and let $e$ and $e'$ be $s$-labelled edges in $G_n$ and $G_{n'}$. Then there is a graph isomorphism between $N_2(e)$ and $N_2(e')$ taking $e$ to $e'$.*

We prove Proposition 11 in Appendix D. This immediately allows us to characterise the balanced Forman curvature and Ollivier curvature for all of the Cayley graphs we generate:

**Proposition 12**  *The balanced Forman curvatures $\mathrm{Ric}(n)$, and the Ollivier curvatures $\kappa(n)$ of all edges of Cayley graphs $G_n$ are given by:*

$$\mathrm{Ric}(n) = \begin{cases} 0 & \text{if } n = 2 \\ -1/4 & \text{if } n = 3 \\ -1/2 & \text{if } n = 4 \\ -1 & \text{if } n \geqslant 5, \end{cases} \qquad \kappa(n) = \begin{cases} 0 & \text{if } n = 2 \\ -1/8 & \text{if } n = 3 \\ -1/4 & \text{if } n = 4 \\ -3/8 & \text{if } n = 5 \\ -1/2 & \text{if } n \geqslant 6. \end{cases}$$

**Proof**  Proposition 11 implies that the balanced Forman and Ollivier curvatures are all equal for $n > 18$. Their values for $2 \leqslant n \leqslant 19$ can all be empirically computed, and are given as above. ∎

## Appendix D.  Proof of Proposition 11

*Let s be one of*

$$\begin{pmatrix} 1 & 1 \\ 0 & 1 \end{pmatrix}, \qquad \begin{pmatrix} 1 & 0 \\ 1 & 1 \end{pmatrix}.$$

*Let $n, n' > 18$ and let $e$ and $e'$ be $s$-labelled edges in $G_n$ and $G_{n'}$. Then there is a graph isomorphism between $N_2(e)$ and $N_2(e')$ taking $e$ to $e'$.*

**Proof**  Note first that, by the homogeneity of the Cayley graphs $G_n$ and $G_{n'}$, we may assume that $e$ and $e'$ emanate from the identity vertex of each graph.

Let $G_\infty$ be the Cayley graph of $\mathrm{SL}(2, \mathbb{Z})$ with respect to the generators

$$S_\infty = \left\{ \begin{pmatrix} 1 & 1 \\ 0 & 1 \end{pmatrix}, \begin{pmatrix} 1 & 0 \\ 1 & 1 \end{pmatrix} \right\}.$$

Let $e_\infty$ be the $s$-labelled edge emanating from the identity vertex of $G_\infty$. The quotient homomorphism

$$\mathrm{SL}(2, \mathbb{Z}) \to \mathrm{SL}(2, \mathbb{Z}_n)$$

induces a graph homomorphism $G_\infty \to G_n$ sending $e_\infty$ to $e$. We will show that it restricts to a graph isomorphism

$$N_2(e_\infty) \to N_2(e).$$

# Extended Abstract Track

As there is a similar graph isomorphism $N_2(e_\infty) \to N_2(e')$, the proposition will follow.

Note that two elements of $\mathrm{SL}(2, \mathbb{Z})$ map to the same element of $\mathrm{SL}(2, \mathbb{Z}_n)$ if and only if they differ by multiplication by an element of the kernel $K_n$. This is

$$K_n = \left\{ \begin{pmatrix} a & b \\ c & d \end{pmatrix} \in \mathrm{SL}(2, \mathbb{Z}) : a \equiv d \equiv 1 \bmod n \text{ and } b \equiv c \equiv 0 \bmod n \right\}.$$

The graph homomorphism sends edges to edges, and so it is distance non-increasing. Hence it certainly sends $N_2(e_\infty)$ to $N_2(e)$. It is also clearly surjective, because any element of $N_2(e)$ is reached from an endpoint of $e$ by a path of length at most 2, and there is a corresponding path in $N_2(e_\infty)$.

We just need to show that this is an injection. If not, then two distinct vertices $g_1$ and $g_2$ in $N_2(e_\infty)$ map to the same vertex in $N_2(e)$. Note then that as elements of $\mathrm{SL}(2, \mathbb{Z})$, $g_2 = g_1 k$ for some $k \in K_n$. There are paths with length at most 3 joining the identity 1 to $g_1$ and $g_2$ respectively. Hence, the distance in $G_\infty$ between $g_1$ and $g_2$ is at most 6. Therefore, the distance between 1 and $g_1^{-1} g_2$ is at most 6. This element $g_1^{-1} g_2$ lies in $K_n$. We will show that when $n > 18$, the only element of $K_n$ that has distance at most 6 from the identity is the identity itself. This will imply that $g_1^{-1} g_2 = 1$ and hence $g_1 = g_2$. But this contradicts the assumption that $g_1$ and $g_2$ are distinct vertices. Our argument follows that of Margulis (1982).

The operator norm $||A||$ of a matrix $A \in \mathrm{SL}(2, \mathbb{Z})$ is

$$||A|| = \sup\{|A(v)| : v \in \mathbb{R}^2, |v| = 1\}.$$

This is submultiplicative: $||AB|| \leqslant ||A|| \, ||B||$ for matrices $A$ and $B$. It can be calculated as the square root of the largest eigenvalue of $A^t A$. In our case, the operator norms satisfy

$$\left\| \begin{pmatrix} 1 & 1 \\ 0 & 1 \end{pmatrix} \right\| = \left\| \begin{pmatrix} 1 & 0 \\ 1 & 1 \end{pmatrix} \right\| = \frac{1 + \sqrt{5}}{2}.$$

Consider an element

$$K = \begin{pmatrix} a & b \\ c & d \end{pmatrix}$$

of $K_n$ that is not the identity. Since $a \equiv d \equiv 1$ modulo $n$ and $b \equiv c \equiv 0$ modulo $n$, we deduce that at least one $|a|$, $|b|$, $|c|$ and $|d|$ is at least $n - 1$. Therefore, this matrix acts on one of the vectors $(1, 0)^t$ or $(0, 1)^t$ by scaling its length by at least $n - 1$. Therefore, $||K|| \geqslant n - 1$. Suppose now that $K$ has distance at most 6 from the identity. Then $K$ can be written as a word in the generators of $\mathrm{SL}(2, \mathbb{Z})$ with length at most 6. Therefore, we obtain the inequality

$$||K|| \leqslant \left( \frac{1 + \sqrt{5}}{2} \right)^6 < 17.95.$$

Hence, $n < 18.95$ and therefore, as $n$ is integral, $n \leqslant 18$. ∎

# Extended Abstract Track

## Appendix E. Proof of Theorem 7

*For any $\delta > 0$ and $\Delta > 0$, there are only finitely many graphs with maximum vertex degree $\Delta$, Cheeger constant at least $\delta$ and non-negative Ollivier curvature.*

**Proof** This is a consequence of the main result of Salez (Salez, 2021, Theorem 3). This states if $G_n = (V_n, E_n)$ is a sequence of graphs with the following properties:

$$\sup_{n \geqslant 1} \left\{ \frac{1}{|V_n|} \sum_{v \in V_n} \deg(v) \log \deg(v) \right\} < \infty \tag{2}$$

$$\forall \epsilon > 0, \quad \frac{1}{|E_n|} |\{e \in E_n : \kappa(e) < -\epsilon\}| \to 0 \text{ as } n \to \infty, \tag{3}$$

then

$$\forall \rho < 1, \quad \liminf_{n \to \infty} \left\{ \frac{1}{|V_n|} |\{i : \mu_i(G_n) \geqslant \rho\}| \right\} > 0.$$

Here, $\kappa(e)$ is the Ollivier curvature of an edge $e$ and

$$1 = \mu_0(G) \geqslant \mu_1(G) \geqslant \cdots \geqslant 0$$

are the eigenvalues of the lazy random walk operator. To prove the theorem, we suppose that on the contrary, there are infinitely many distinct graphs $G_n = (V_n, E_n)$ with maximum vertex degree $\Delta$, Cheeger constant at least $\delta$ and non-negative Olliver curvature. Then

$$\sum_{v \in V_n} \deg(v) \log \deg(v) \leqslant |V_n| \Delta \log \Delta$$

and so condition 2 is satsfied. Condition 3 is trivially satisfied because the Ollivier curvature of each graph is non-negative. Thus, we deduce that the conclusion of Salez' theorem holds. Setting $\rho = 1 - (\delta^2/4\Delta^2)$, we deduce that a definite proportion of the eigenvalues of the lazy random walk operator are at least $1 - (\delta^2/4\Delta^2)$. In particular, $\mu_1(G_n) \geqslant 1 - (\delta^2/4\Delta^2)$. Denote the eigenvalues of the normalised Laplacian by

$$0 = \lambda'_0(G_n) \leqslant \lambda'_1(G_n) \leqslant \ldots$$

These are related to the eigenvalues of the lazy random walk operator by $\lambda'_i(G_n) = 2 - 2\mu_i(G_n)$. Hence, $\lambda'_1(G_n) \leqslant \delta^2/(2\Delta^2)$. There is a variation of Cheeger's inequality that relates $\lambda'_1$ to the *conductance* of the graph. To define this, one considers subsets $A$ of the vertex set, and defines their *volume* to be $\mathrm{vol}(A) = \sum_{v \in A} \deg(v)$. The conductance $\phi(G)$ of a graph $G$ is

$$\phi(G) = \min \left\{ \frac{|\partial A|}{\mathrm{vol}(A)} : A \subset V(G), \, 0 < \mathrm{vol}(A) \leqslant \mathrm{vol}(V(G))/2 \right\}.$$

Then, by Chung (1997, Theorem 2.2),

$$\phi(G) \leqslant \sqrt{2\lambda'_1(G)}$$

Hence, in our case,

$$\phi(G_n) \leqslant \delta/\Delta.$$

*Extended Abstract Track*

Consider any subset $A_n$ of the vertex set that realises $\phi(G_n)$. Thus $0 < \text{vol}(A_n) \leqslant \text{vol}(V_n)/2$ and $|\partial A_n|/\text{vol}(A_n) = \phi(G_n) \leqslant \delta/\Delta$. If $A_n$ is at most half the vertices of $G_n$, then this implies that the Cheeger constant $h(G_n) \leqslant \delta$. On the other hand, if $A_n$ is more than half the vertices of $G_n$, we consider its complement $A_n^c$. Its cardinality $|A_n^c|$ satisfies

$$|A_n^c| \geqslant \text{vol}(A_n^c)/\Delta.$$

Hence,

$$h(G_n) \leqslant \frac{|\partial A_n^c|}{|A_n^c|} \leqslant \frac{|\partial A_n|\Delta}{\text{vol}(A_n^c)} \leqslant \frac{|\partial A_n|\Delta}{\text{vol}(A_n)} = \phi(G_n)\Delta \leqslant \delta.$$

In either case, we deduce that the Cheeger constant of $G_n$ is at most $\delta$, contradicting one of our hypotheses. Hence, there must have been only finitely many graphs satisfying the conditions of the theorem. ∎

## Appendix F. Cayley graphs are locally 'tree-like'

The negative curvature of each edge in $G_n$ implies that they are locally 'tree-like'. In Appendix G, we make this statement precise by showing that $G_n$ is 'tree-like' up to scale $c\log(n)$ about each node, for $c \simeq (1/2)(\log((1 + \sqrt{5})/2))^{-1}$ (see Figure 1 (Right) for a schematic view).

This tree-like structure might seem, at first, to be counter-productive for good propagation across the graphs $G_n$. Indeed, GNNs based on trees have been shown to have provably poor performance (Alon and Yahav, 2020). The reason for this seems to be two-fold. On the one hand, trees have small Cheeger constant. Indeed, any tree $G$ on $n$ vertices has a Cheeger constant $1/\lfloor n/2 \rfloor$, since we may find an edge that, when removed, decomposes the graph into subgraphs with $\lfloor n/2 \rfloor$ and $\lceil n/2 \rceil$ vertices. As discussed in Section 2 and in Topping et al. (2021), when a graph has small Cheeger constant, its performance when used as a template for a GNN is likely to become poor. Secondly, GNNs based on trees are susceptible to oversquashing. For a $k$-regular infinite tree, there are $k(k-1)^{r-1}$ vertices at distance $r$ from a given vertex. Hence, if information is to be propagated at least distance $r$ from a given vertex, then seemingly an exponential amount of information is required to be stored.

However, neither of these issues are problematic for a GNN based on the Cayley graph $G_n$. By Theorem 6, their Cheeger constants are bounded away from 0. Secondly, although they are tree-like locally, this is only true up to scale $O(\log n)$. In fact, the $r$-neighbourhood of any vertex is the whole graph $G_n$ as soon as $r > C\log n$, for some constant $C$, by Theorem 4. Being tree-like up to distance $O(\log n)$ does not lead to a requirement to store too much information as the message propagates. This is because $k(k-1)^{r-1}$ is polynomial in $n$ when $r \leqslant O(\log n)$.

## Appendix G. Cayley graph at infinity is quasi-isometric to a tree

As all vertices of $G_n$ look the same, we focus attention on $N_r(1)$, the $r$-neighbourhood of the identity vertex. The proof of Proposition 11 immediately gives the following.

# Extended Abstract Track

**Proposition 13** *Let $r$ be a positive integer satisfying*

$$r < \frac{1}{2}\left(\log\left(\frac{1+\sqrt{5}}{2}\right)\right)^{-1}\log(n-1).$$

*Then there is a graph isomorphism between the $r$-neighbourhood of the identity vertex in $G_n$ and the $r$-neighbourhood of the identity vertex in $G_\infty$. This isomorphism takes the identity vertex to the identity vertex.*

**Proof** As shown in the proof of Proposition 11, there is a graph homomorphsm from $N_r(1)$ in $G_\infty$ to $N_r(1)$ in $G_n$ that is a surjection. If it fails to be an injection, then there is a non-trivial element $K$ in the kernel $K_n$ of $\mathrm{SL}(2,\mathbb{Z}) \to \mathrm{SL}(2,\mathbb{Z}_n)$ satisfying

$$||K|| \leqslant \left(\frac{1+\sqrt{5}}{2}\right)^{2r}.$$

But any non-trivial element $K$ in $K_n$ satisfies

$$||K|| \geqslant n-1.$$

Rearranging gives the required inequality. ∎

This raises the question of the local structure of $G_\infty$. The answer is well-known: it is 'tree-like'. Specifically, it is quasi-isometric to a tree. The formal definition of quasi-isometry is as follows.

**Definition 14** *A* quasi-isometry *between two metric spaces $(X_1, d_1)$ and $(X_2, d_2)$ is a function $f\colon X_1 \to X_2$ that satisfies the following two conditions:*

*1. there are constants $c, C > 0$ such that, for every $x, x' \in X_1$*

$$c\,d_1(x,x') - c \leqslant d_2(f(x), f(x')) \leqslant C\,d_1(x,x') + C,$$

*2. there is a constant $K \geqslant 0$ such that for every $y \in X_2$, there is an $x \in X_1$ with $d_2(f(x), y) \leqslant K$.*

*If there is such a quasi-isometry, we say that $(X_1, d_1)$ and $(X_2, d_2)$ are* quasi-isometric.

This forms an equivalence relation on metric spaces. When two metric spaces are quasi-isometric, they are viewed as being 'essentially the same' at large scales.

When $S$ and $S'$ are finite generating sets for a group $\Gamma$, the graphs $\mathrm{Cay}(\Gamma; S)$ and $\mathrm{Cay}(\Gamma; S')$ are quasi-isometric. Hence, the quasi-isometry type of a finitely generated group is well-defined, and this is the central object of study in geometric group theory.

The group $\mathrm{SL}(2,\mathbb{Z})$ has a finite-index subgroup that is a free group $F$ (Trees, 2003). If $S'$ denotes a free generating set for $F$, then $\mathrm{Cay}(F; S')$ is a tree. As passing to a finite-index subgroup preserves its quasi-isometry class, we deduce that the Cayley graph $G_\infty = \mathrm{Cay}(\mathrm{SL}(2,\mathbb{Z}); S_\infty))$ is indeed quasi-isometric to a tree, as claimed above.

Extended Abstract Track

## Appendix H. Mixing time properties of expander graphs

Expanders are well known to have small mixing time, in the following sense.

Let $G$ be a graph. We will consider probability distributions $\pi$ on $V(G)$. The lazy random walk operator $M$ acts on probability distributions as follows. We think of $\pi(v)$ as being the probability of the random walk being at vertex $v$. If the current location of the walk is at $v$, then at the next step of the walk, either we stay put with probability $1/2$ or we move to one of its neighbours with equal probability. Then $M\pi$ is the new probability distribution.

In the case when $G$ is $k$-regular, this takes a particular simple form. The operator $M$ is represented by the matrix $(1/2)I + (1/2k)A$, where $A$ is the adjacency matrix. In that case, any initial distribution $\pi$ converges under powers of $M$ to the uniform distribution.

This is true for any reasonable notion of convergence, but we will use the $\|\cdot\|_1$ norm, where for two probability distributions $\pi$ and $\pi'$,

$$\left\|\pi - \pi'\right\|_1 = \sum_{v \in V(G)} |\pi(v) - \pi'(v)|.$$

**Definition 15** *The* mixing time *for a regular graph $G$ is the minimum value of $\ell$ such that for any starting probability distribution $\pi$ on the vertex set of $G$,*

$$\left\|M^\ell \pi - u\right\|_1 \leqslant \frac{1}{4}.$$

*Here, $u$ is the uniform probability distribution on the vertex set, and $M$ is the lazy random walk operator.*

Expanders have small mixing times in the following very strong sense.

**Theorem 16** *For any $k > 0$ and $\delta > 0$, there is a constant $c > 0$ with the following property. If $G$ is a connected $k$-regular graph on $n$ vertices with Cheeger constant at least $\delta > 0$, then the mixing time for $G$ is at most $c \log(n)$.*

## Appendix I. EGP forward pass pseudocode

In Algorithm 1, we provide the forward pass for our EGP layer, linking it back to previous equations.

# Extended Abstract Track

**Algorithm 1:** Expander graph propagation (EGP) forward pass

**Inputs :** Node features $\mathbf{X} \in \mathbb{R}^{|V| \times k}$, Adjacency matrix $\mathbf{A} \in \mathbb{R}^{|V| \times |V|}$

**Output :** Node embeddings $\mathbf{H}$

```
// Choose the smallest Cayley graph from our family that has number of
   nodes equal to, or greater than, |V|
```
$n \leftarrow \operatorname{argmin}_{m \in \mathbb{N}} |V(\operatorname{Cay}(\operatorname{SL}(2, \mathbb{Z}_m); S_m))| \geqslant |V|$

$G^{\operatorname{Cay}(n)} \leftarrow \operatorname{Cay}(\operatorname{SL}(2, \mathbb{Z}_n); S_n)$

$\mathbf{A}_{uv}^{\operatorname{Cay}(n)} \leftarrow \begin{cases} 1 & (u,v) \in E(G^{\operatorname{Cay}(n)}) \\ 0 & \text{otherwise} \end{cases}$ ;    `// Populate adjacency matrix of the Cayley`
`graph`

$\mathbf{H}^{(0)} \leftarrow \mathbf{X}$;                                   `// Initialise GNN inputs`

**for** $t \in \{1, \ldots, T\}$ **do**

  **if** $t \mod 2 = 0$ **then**

    $\mathbf{H}^{(t)} \leftarrow \operatorname{GNN}^{(t)}(\mathbf{H}^{(t-1)}, \mathbf{A})$ ; `// GNN layer over input graph; e.g. Equation`
    1

  **end**

  **else**

    $\mathbf{H}^{(t)} \leftarrow \operatorname{GNN}^{\operatorname{Cay}}\left(\mathbf{H}^{(t-1)}, \mathbf{A}_{1:|V|,1:|V|}^{\operatorname{Cay}(n)}\right)$;   `// GNN layer over Cayley graph; e.g.`
    Equation 1

  **end**

**end**

**return** $\mathbf{H}^{(T)}$ ;                      `// Return final embeddings for downstream use`

