# OpenReview forum: "Expander Graph Propagation"
_NeurIPS.cc/2022/Workshop/NeurReps — NeurReps 2022 Oral_

### Official Review · Reviewer_2Gja · 2022-10-10
**Review of Expander Graph Propagation**

**Confidence:** 5
**Soundness:** 4
**Presentation:** 4
**Contribution:** 3
**Overall Rating:** 7

**Summary:**

The paper proposes an efficient method for alleviating the over-squashing effects in GNNs by interleaving a GNN layer on the given graph input topology with a GNN layer over a Cayley graph of suitable dimension.

**Questions:**

It would be relevant to analyse a little the stability of this process. How sensitive we are to the depth (number of layers on the Cayley graph)? This question would also be important into addressing the issue of whether the improvement over baselines is truly deriving from long-range dependencies propagating better on the Cayley graph.

**Limitations:**

Yes.

**Recommended Decision:**

3: Accept

**Relevance:**

4: Highly relevant

**Strengths And Weaknesses:**

The main idea is interesting given the many desirable properties of expander graphs and will indeed avoid pre-computations (rewiring operations), be efficient and alleviate the over-squashing effect by `improving' the Cheeger constant. Empirical evaluation -- albeit not the most extensive one -- is also encouraging.

Weaknesses: It is not entirely clear what we lose by using a graph independent from topology and data to propagate messages and whether this is the most efficient way of alleviating the over-squashing. Indeed, the comparison with curvature only makes sense on a global scale but in general it is possible to have sparsity and positive curvature in "some" regions of the graph.

**Submission Track:**

Extended Abstract (4 Page)

---

### Official Review · Reviewer_LDRb · 2022-10-14
**Review of Expander Graph Propogation**

**Confidence:** 3
**Soundness:** 3
**Presentation:** 3
**Contribution:** 3
**Overall Rating:** 8

**Summary:**

This paper studies the use of expander graphs in Graph Neural Networks (GNN). One would ideally seek methods that are capable of propagating information globally in a graph, that are protected from oversquashing, that are computationally efficient (subquadratic) in time and space, and that do not require dedicated processing of the input. The paper gives a thorough and well-explained introduction to expander graphs, to the eigenvalues of their Laplacians, to the Cheeger constant of a graph, to the diameter of a graph, and to the relationship between these concepts. The authors then propose expander graphs for use in GNNs in order to make steps towards the four ideal method properties mentioned above.

**Questions:**

Is there space in the paper, or in the appendix, to explain the over-squashing property more?

In the abstract you write "we theoretically demonstrate that ... are likely", and on the last sentence of the first page you write "We ... prove that the existence of negatively curved edges ... might in fact be required." I was surprised to see the soft words "are likely" or "might" in the description of a theoretical result or a proof. Why is this?

**Limitations:**

This is a theoretical paper without experimental tests on applied problems.

**Recommended Decision:**

3: Accept

**Relevance:**

3: Solid fit

**Strengths And Weaknesses:**

This paper is a well-explained introduction to the use of expander graphs in graphical neural networks (GNNs). The paper is clear and presented in an engaging manner. The results are precise and well-motivated. However, a downside is that the paper is mostly theoretical, and does not test the use of expander graphs in the application of GNNs to practical problems.

**Submission Track:**

Extended Abstract (4 Page)

---

### Official Review · Reviewer_isXN · 2022-10-15
**Theoretical work on Expander Graph Propagation**

**Confidence:** 4
**Soundness:** 4
**Presentation:** 4
**Contribution:** 4
**Overall Rating:** 9

**Summary:**

The paper proposes a way so that the bottlenecks and over-squashing in the existing GNNs can be avoided. This is done by the addition of an extra GNN layer over the Cayley graph after a GNN layer on the input graph.

**Questions:**

1. While the paper gives a very detailed theoretical background, it would also be useful if it could provide an overview of more experimental results.
2. For which other tasks than graph classification can the method be used? Are the experimental evidence available for those tasks as well?

**Limitations:**

Experimental results for other tasks could be shown. However it is a very well written paper.

**Recommended Decision:**

3: Accept

**Relevance:**

3: Solid fit

**Strengths And Weaknesses:**

The work is novel and interesting. A thorough theoretical analysis is done that supports the claims made. The submission is clearly written and well organized. The results look interesting and can be used to overcome the problems of bottlenecks and over squashing in GNNs architectures.

**Submission Track:**

Extended Abstract (4 Page)

---

### Decision · Program_Chairs · 2022-10-21

Accept (Oral)